# Spatial–Temporal Patterns of Population Aging in Rural China

**DOI:** 10.3390/ijerph192315631

**Published:** 2022-11-24

**Authors:** Chan Chen, Jie Li, Jian Huang

**Affiliations:** School of Geographical Science, Guangzhou University, Guangzhou 510006, China

**Keywords:** population aging, rural, China, Geographical Detector, county level

## Abstract

(1) Background: Population aging has been accelerating in China since the 1990s, and the number of people over 65 reached 190 million in 2020. However, the spatial distribution of the aged is not homogeneous; in rural areas, the aged population accounted for 17.72% of the total population, whereas in urban areas, it accounted for 11.11%, which is 6.61 p.p. less. Therefore, this study aims to examine the spatial heterogeneity and influencing factors of population aging in rural China from 2000 to 2020. (2) Methods: First, Getis–Ord Gi* was used to analyze the spatial clustering of the aged population in rural China. Then, standard deviational ellipse was used to characterize the temporal trend of the spatial clustering of population aging in rural China. Finally, potential influencing factors that could have contributed to the spatial–temporal patterns were analyzed using a novel spatial statistical package “Geographical Detector”. (3) Results: (a). Aging in rural populations increased and occurred throughout China from 2000 to 2020. (b). The spatial patterns of aging in China are roughly divided by the Hu Line, which is the population density boundary of China. (c). The mean center of the aged population tended to orient around a northeast-to-southwest major axis over the past 20 years, contrary to the internal migration pattern that flows from north to south. (d). The population age structure, longevity rate, and fertility rate were the predominant factors of aging in rural areas. (4) Conclusions: As the aged population is rapidly increasing in rural areas in China in a spatially heterogeneous fashion, governments and private sectors need to collaborate to alleviate the problem.

## 1. Introduction

Population aging is becoming one of the most crucial social transformations of the 21st century and is being given close attention and more importance worldwide. As noted by the World Population Prospects 2019, one in six people in the world will be over the age of 65 by 2050 (16%), up from one in eleven in 2019 (9%) [1]. The world population is growing older, while the growth varies greatly across regions [2]. Take China as an example. The number of the aged population increased from 88.21 to 102.4 million from 2000 to 2020. By the end of 2020, China had a population of just over 1.41 billion, of which 0.191 billion were over 65 years old, accounting for 13.5% of the total population [3]. China’s population is projected to peak at about 1.35 to 1.46 billion in 2035 and will then continue to decline until 2050, but the trend of population aging will continue.

In particular, the aged population is distributed at different levels in rural and urban areas in China. In rural areas, population aging is more severe than that in urban areas, and the problem is aggravated by generally less developed infrastructures and fewer healthcare facilities in rural areas. In pursuit of economic profit and personal development, many people in rural areas migrate to “big cities”, leaving the old (as well as the young) in their rural homes, creating the so-called “empty hearted village” and “empty nester family” phenomenon. Due to the limitation of urban–rural dualism, migrant workers do not have the same welfare, healthcare protection, or education opportunities for their children as their urban counterparts. As a result, population aging in China has a different pattern in rural and urban areas. Social, economic, and health problems associated with the left-behind elderly in rural China need to be dealt with scientifically and sustainably to improve the well-being of all.

Over the past 40 years, the studies on China’s population aging have always been a focus for the scientific community and governmental organizations. Traditionally, studies on population aging in China mostly focus on analyzing the current situation and development trends of China’s population aging and point out that China’s population aging presents significant characteristics such as a large-scale elderly population, fast growth rate, uneven distribution of aging, and “getting old before getting rich” [4]. Due to the vigorous development of geographic information technology in the 1990s, the spatial analysis method is widely applied to the study of the aging population, such as Global Moran’s I and kernel density analysis, and considerable literature has grown up around the theme of the spatial and temporal characteristics of the aging population distribution in recent decades [5,6]. In addition, the research to date has fully described the influencing factors of population aging, including the impact of aging on society, economy, urbanization, and rural development, as well as the natural environmental factors [7,8]. However, existing research mainly focuses on qualitative studies on the overall pattern of aging, the general policy on population aging, and the differences between rural and urban areas in China; the combined research on the spatiotemporal pattern, especially the differences in aging between rural and urban areas and its potential influencing factors are still in their early stages. Moreover, most research pays less attention to rural areas. Considering the importance of how continued population aging in the coming decades influences the sustainability of rural regions and positive aging welfare, it is necessary to analyze the spatial patterns of rural aging in China with the newest population census data in 2020.

This study set out to examine the spatial heterogeneity and influencing factors of population aging in rural China from 2000 to 2020 using county-level panel data. Based on a large spatial scale and time span, we comprehensively used GIS analysis tools to study the spatial and temporal distribution law and influencing factors of China’s rural population aging. The aims of this study were to reveal the comprehensive deepening law of rural population aging and provide a scientific basis for active aging development and population policy adjustment in China.

## 2. Materials

In this part, we define the study area, and the indicators for evaluating the degree of aging of the rural population are described. Then, a system of variables influencing the aging of rural population is shared, and the reasons for selecting these variables are explained. Finally, the data sources used are described.

### 2.1. Study Area

Administrative divisions are areas that are divided hierarchically to facilitate administration by the country. According to the Constitution of the People’s Republic of China, the administrative regions are divided into four levels: provincial, prefecture, county, and township. Currently, there are 34 provincial-level administrative regions (including 23 provinces, 5 autonomous regions, 4 municipalities directly under the central Government, and 2 special administrative regions), 333 prefecture-level administrative regions, 2846 county-level administrative regions, and 38,755 township-level administrative regions in China. Among them, townships, i.e., rural areas, are the most basic administrative units [9]. Furthermore, to reflect the social and economic development of various zones, administrative units on mainland China are divided into seven areas with similar economic development levels, and relatively uniform geographic locations. These are Central China, Eastern China, Northern China, Northeast China, Northwest China, Southern China, and Southwest China [10] (Figure 1). As the statistical data of township administrative districts is insufficient and difficult to obtain, this paper studies the rural population aging in China based on the county-level administrative units (2846 in total). In addition, as there is no data for Hong Kong, Macau, and Taiwan, they are excluded from this study.

### 2.2. Evaluation Index of Rural Population Aging

There are various measures of population aging, including the ratio of the elderly population, the ratio of the elderly to the young, the dependency ratio of the elderly population, and the median age of the population, among which the proportion of the elderly population in total is the most used measure. According to the aging standard of the United Nations in 1956, when the ratio of old people to total is greater than or equal to 7%, it is considered to be entering an aging society; if it is greater than 14%, it is considered as a hyper-aged society [11]. Therefore, this paper defines the proportion of the population aged 65 and over in the total population as an index to measure the degree of rural population aging (RPA) in each county.

In order to better compare the degree of aging in different regions, this paper refers to the international standards and divides RPA into three different aging classes. If the RPA is less than or equal to 7%, it is considered a young type of rural; if the RPA is greater than or less than 14%, it is named an aged type of rural; if the RPA is greater than 14%, it is called a hyper-aged type of rural.

### 2.3. Construction of the Indicator System

Aging is essentially a special population age structure, and the age change in the population plays a decisive role in its development and evolution. In addition, aging is also affected by the local population base, changes in population mechanical growth, economic development status, healthcare accessibility, and education background. Due to the complex and diverse topography, climate, and hydrological conditions in China, there are large differences in population and economic development between the east and the west. Therefore, this paper constructs an influential factor system of rural aging through the four levels of population, socioeconomics, healthcare, and education to explore the influencing factors affecting the spatiotemporal pattern of rural population aging in China. Table 1 shows the selection and description of the variables of each system.

In terms of the demographic characteristics, we selected the total population (TPOP) of each county, proportion of the population aged 55–64 (P55-64) ten years ago, fertility rate (FER), longevity rate (LGV), and migration rate (MIG) to characterize the natural and mechanical population change. The TPOP reflects the population size of an area. The P55-64 in each county in 1990, 2000, and 2010 refers to the population base of the elderly. The higher the P55-64, the greater the impact on population aging changes after 10 years [12]. The FER refers to the proportion of the new generation population, which is considered one of the mechanisms underlying demographic change [13]. The trend of low fertility rates and a reduction in the size of the birth population will further exacerbate population aging. The LGV is the proportion of people aged 80 and over in the total, as people over 80 are regarded as having longevity, referring to the classification of the World Health Organization of the United Nations [14]. One of the best achievements of modern civilization has been the enormous reduction in human mortality [15], and the significant prolongation in life expectancy has had a significant impact on aging.

The urbanization rate (UBZ) refers to the proportion of the nonagricultural registered population in the total population, and the per capita GDP (PGDP) reflects the economic development of a region. Both the UBZ and PGDP have an impact on migration patterns. China implements a dual household registration system; nonagricultural-registered residents enjoy better medical, education, employment, and housing conditions, whereas rural residents do not receive the same benefits [16]. This system promotes the migration of young laborers from rural to urban areas and accelerates urbanization to a great extent [17]. People often leave the countryside to pursue better economic benefits and go to cities with more developed economies and better infrastructure for employment and life. Generally, immigrant parents have relatively low mobility and do not tend to move with their children, which has led to issues with “empty-hearted villages” and “empty nest household” in rural China.

As for the healthcare variables, the number of hospitals (HOS) and beds (BED) reflects the status of the healthcare infrastructure. The improvement of medical and health conditions can prolong life expectancy, thus indirectly affecting the population aging.

The per capita years of education (PEDU) and illiteracy rate (ILT) are two indicators that reflect the educational status from different perspectives. The PEDU is an intensity indicator that reflects the overall level of education, whereas the ILT is a structural indicator that indicates the popularization of education. With the improvement in education levels in China, the return on investment in education has increased, which causes families to pay more attention to the quality of education rather than birth. Women’s intentions to have children are indirectly influenced by the PEDU and ILT, and the proportion of older people increases as the number of births declines.

### 2.4. Data Sources and Arrangement

The RPA, TPOP, and P55-64 data are from the fifth to seventh China population censuses in 2000, 2010, and 2020, respectively [18,19,20]. As some of the counties have not publish the seventh national census data yet, we used the prefecture-level data instead. A population census in China is conducted every ten years. The relevant departments in all regions conduct a comprehensive survey and registration of the existing population of the country on a general, household-by-person basis in strict accordance with the instructions and laws. The focus of the census is to master, analyze, and predict the development and changes of the existing population in various places, mainly to understand the sex ratio, sex ratio at birth, single and married populations, elderly population, etc. The annual data of the fertility rate (FER), longevity rate (LGV), migration rate (MIG), urbanization rate (UBZ), per capita GDP (PGDP), per capita education (PEDU), illiteracy rate (ILT), and the number of hospitals (HOS) and beds (BED) can be collected in the *Statistic Yearbook* published by the National Bureau of Statistics of China [3].

## 3. Methods

In this study, we focused on examining the spatial distribution and evolution patterns and revealing the influential factors of RPA. First, we utilized the Getis–Ord Gi* method to analyze the spatial distribution patterns of rural population aging. Then, from a dynamic perspective, the standard deviational ellipse method was applied to analyze the change in the distribution range and the trajectory of gravity. Finally, we analyzed the potential driving factors of the spatial patterns from demographic, socioeconomic, healthcare, and educational perspectives using the Geographical Detector method.

### 3.1. Getis–Ord Gi*

The Getis–Ord Gi* (pronounced G-i-star) method, originally developed by Getis and Ord [21], is used to identify a tendency for positive spatial clustering and can distinguish high and low spatial associations between the locations [22]. In this study, Getis–Ord Gi* was used to detect the high and low value clustering areas of RPA. A simple form of Gi* statistics is:(1)$$G_{i}{}^{*}=\frac{\sum_{j=1}^{n}W_{ij}x_{j}}{\sum_{j}^{n}x_{j}}$$
where the *Gi** statistic describes the spatial dependency of incident i over all n events, $W_{ij}$ is the spatial weight between feature i and j, $x_{j}$ is the attribute value of feature j, n is the total number of features, and $\bar{X}$ is calculated as:(2)$$\bar{X}=\frac{\sum_{j=1}^{n}x_{j}}{n}$$

The $G_{i}{}^{*}$ statistic is expressed in the form of a statistically significant z-score. When the z-score is high, the clustering of the high value will be intense. The resultant z-score indicates whether features with high or low values cluster geographically. A z-score near zero indicates no apparent spatial clustering. This tool works by examining each feature within the context of neighboring features. Getis–Ord Gi* identifies significant spatial clusters of high (hot spots) and low values (cold spots).

### 3.2. Standard Deviational Ellipse

The standard deviational ellipse (SDE), introduced by sociologist Welty Lefever in 1926 [23], is one of the most classical and popular methods to analyze the directional characteristics of spatial distribution [24]. This model creates ellipses or ellipsoids to summarize the spatial characteristics of geographic features, including central tendency, dispersion, and directional trends. It is convinced that an ellipse can geographically represent spatial data in a highly effective way [25]. The spatial–temporal evolution of seasonal tornado activity [26], the spread of the COVID-19 epidemic [27], population distribution [28], and other topics are investigated using the SDE method. The calculation of the SDE includes the center coordinates, azimuth θ, long axis, and short axis. Among them, the center coordinates represent the center of gravity of the space elements. The azimuth θ, which can be defined as the direction of the long axis, determines the orientation of an ellipse [25], whereas the long and short axes determine the shape, indicating the distribution density of a set of geographical units in one- and two-dimensional spaces, respectively [29].

To create the SDE, the initial step is to take the mean of the *x* and *y* coordinates of all of the n units studied to calculate the center coordinates [24]. Then, calculate the azimuth θ. Finally, referring to the calculation formulas of various parameters mentioned in the literature of Lefever, Gong, Yuill, and Xia [23,24,25,29], the formulas are as follows:

Step 1: Calculation of center coordinate of the ellipse:(3)$$SDE_{x}=\sqrt{\frac{\sum_{i=1}^{n}{\left(x_{i}-\bar{x}\right)}^{2}}{n}},\;SDE_{y}=\sqrt{\frac{\sum_{i=1}^{n}{\left(y_{i}-\bar{y}\right)}^{2}}{n}}$$
(4)$$\bar{x}=\frac{1}{n}\overset{n}{\underset{i=1}{\sum }}x_{i},\;\bar{y}=\frac{1}{n}\overset{n}{\underset{i=1}{\sum }}y_{i}$$
where {($x_{i},y_{i});i=1,2,\ldots \ldots ,n\}$ are the coordinates of the geographical units studied. $SDE_{x}$ and $SDE_{y}$ are the *x*, *y* coordinates of the mean center. The mean center coordinates, as the center of the geographical elements, reflect the relative position of rural population aging in a two-dimensional space in this study.

Step 2: Calculate the direction (azimuth *θ*) of the ellipse:(5)$$tan_{\theta }=\frac{\left(\sum_{i=1}^{n}{\overset{‗}{x}}_{i}{}^{2}-\sum_{i=1}^{n}{\overset{‗}{y}}_{i}{}^{2}\right)+\sqrt{{\left(\sum_{i=1}^{n}{\overset{‗}{x}}_{i}{}^{2}-\sum_{i=1}^{n}{\overset{‗}{y}}_{i}{}^{2}\right)}^{2}+4{\left(\sum_{i=1}^{n}{\overset{‗}{x}}_{i}{\overset{‗}{y}}_{i}\right)}^{2}}}{2\sum_{i=1}^{n}{\overset{‗}{x}}_{i}{\overset{‗}{y}}_{i}}$$
where ${\overset{‗}{x}}_{i}$ and ${\overset{‗}{y}}_{i}$ are the deviations between the coordinates of the i point element and average center coordinates.

Step 3: Calculate the long and short axes of the ellipse:(6)$$\alpha_{x}=\sqrt{\frac{2\sum_{i=1}^{n}{\left({\overset{‗}{x}}_{i}\cos\theta -{\overset{‗}{y}}_{i}\sin\theta \right)}^{2}}{n}},\;\alpha_{y}=\sqrt{\frac{2\sum_{i=1}^{n}{\left({\overset{‗}{x}}_{i}\sin\theta +{\overset{‗}{y}}_{i}\cos\theta \right)}^{2}}{n}}$$

In this research, the SDE method was employed to reveal the dynamic changes and spatial evolution of rural aging in China. We used the SDE tool provided by ArcGIS software (Version 10.3, Redlands, CA, USA) to conduct the analysis on the spatial–temporal analysis of population aging in China. The center coordinates reflect the center of gravity of the population aging in counties as well as in rural areas in China. The long axis reflects the main trend of rural aging in one- and two-dimensional space. The short axis, vertical to the long axis, represents the distribution range of rural population aging. The larger the difference between the values of the long and short axes, the more obvious the direction of rural aging. The shorter the short axis is, the stronger the centripetal force is. The azimuth is the rotation angle between the long axis and the north direction clockwise and indicates the distribution direction of each county in China.

### 3.3. Geographical Detector: Influence Factor Analysis of Rural Population Aging

In this research, we used the model to detect the spatial stratified heterogeneity, which is one basic characteristic of geographic phenomena. To explore the determinant power and interactive impact of related factors on rural population aging, the Geographical Detector (Geodetector, GD) model was employed in this study.

The GD model, originally proposed by Wang and Hu, serves as an effective spatial statistics method based on the spatial variation analysis of the geographical strata of variables [30,31]. The GD model has been extensively used to identify the driving factors in a variety of fields, such as public health, environmental pollution, land use, urban livability, and population distribution [32,33,34,35]. The GD model includes four detector modules: factor, interaction, ecological, and risk. The mathematical expressions are as follows:(1)The factor detector quantifies the influences of factors on the *q-statistics*. In this study, the factor detector identifies which factors are responsible for the RPA. Its formula is:(7)$$q=1-\frac{\sum_{h=1}^{L}N_{h}\sigma_{h}{}^{2}}{N_{\sigma }{}^{2}}=1-\frac{SSW}{SST}$$
where q is the explanatory power of the determinants associated with the RPA. $\left\{\;h=1,\;\ldots ,\;L\right\}$ are the stratification of y or factor x, i.e., classification or partition; $N_{h}$ and N represent the number of units in h and the whole region, respectively. $\sigma_{h}{}^{2}$ and $\sigma^{2}$ are the variance of units in h and the global variance of y over the whole region, respectively. SSW indicates the sum of squares, whereas SST represents the total sum of squares. The *q-statistics* range from 0 to 1; the larger the *q-statistics* are, the stronger the influence of factor is.(2)The interaction detector examines whether two independent variables, when taken together, weaken or enhance each another or whether they are independent in developing dependent variables [36]. In this study, the interaction detector examines whether the factors ($x_{1}$ and $x_{2}$) have an interactive effect on RPA. First, the *q-statistics* of factors $x_{1}$ and $x_{2}$, in respect of the RPA, were calculated and marked as $q(x_{1})$ and $q(x_{2})$. Then, the interactive *q-statistics* of factors $x_{1}$ and $x_{2}$ were calculated and marked as $q(x_{1}\cap x_{2})$. The interactive relationship can be classified into five types by comparing the interactive *q-statistics* of the two factors and the *q-statistics* of each of the two factors [37]. The five types are described in Table 2.(3)The ecological detector, which is determined by the F-statistics, is used to compare whether the impacts of the two factors ($x_{1}$ and $x_{2}$) on the dependent variable have a significant difference [38]:(8)$$F=\frac{N_{x_{1}}\left(N_{x_{2}}-1\right)SSW_{x_{1}}}{N_{x_{2}}\left(N_{x_{1}}-1\right)SSW_{x_{2}}}$$
(9)$$SSW_{x_{1}}=\sum_{h=1}^{L_{1}}N_{h}\sigma_{h}{}^{2},\;SSW_{x_{2}}=\sum_{h=2}^{L_{2}}N_{h}\sigma_{h}{}^{2}$$
where $N_{x_{1}}$ and $N_{x_{2}}$ mean the sample number of factors $x_{1}$ and $x_{2}$, respectively. $SSW_{x_{1}}$ and $SSW_{x_{2}}$, respectively, denote the sum of the within-strata variances formed by $x_{1}$ and $x_{2}$. $L_{1}$ and $L_{2}$ represent the number of stratifications of factors $x_{1}$ and $x_{2}$, respectively. If the null hypothesis $H_{0}$: $SSW_{x_{1}}=SSW_{x_{2}}$ is rejected at the confidence level α (usually 5%), the influences of $x_{1}$ and $x_{2}$ on the dependent variable are statistically significant. That is to say, the effect of factor $x_{1}$ on RPA is significantly different from that of factor $x_{2}$ [39].(4)The risk detector is used to detect whether the spatial–temporal pattern of RPA is remarkably different, whereas the area studied is stratified by a variety of factors. If the result of the two factors is “Y”, it means there are significant differences between the two factors that influence RPA, whereas if the result of the two factors is “N”, it means there is no significant difference. The risk detection is examined using *t-statistics*:(10)t=Y¯h=1−Y¯h=2VarYh=1nh=1+VarYh=2nh=212
where ${\bar{Y}}_{h}$ denotes the average of Y in the subregion h, $n_{h}$ represents the size of samples in the subregion h, and Var is variance.


## 4. Results

### 4.1. Spatial Distributions of Rural Aging in China

As mentioned in Section 2.2, the RPA (the percentage of rural population aged 65 and over) was utilized in this study as a measure of rural population aging. To better compare the differences in RPA across regions in China, RPA was divided into three classes: young type if the RPA is below 7%, aged type if the RPA is between 7% and 14%, and hyper-aged type if the RPA is above 14%. Referring to this classification, a map of the spatial distribution profiles of China’s rural population aging ratio from 2000 to 2020 was drawn (see Figure 2a–c).

In 2000, the rural population aging in China was dominated by the young and aged types. Notably, 53.2% of the units were of the young type and were widely distributed across 31 provinces in China. Except for Tibet, Xinjiang, and Ningxia, 46.62% of the units in 28 provinces were of the aged type. In addition, more than 50% of units were the aged type in 14 provinces, including Zhejiang, Jiangsu, Anhui, Chongqing, Shandong, Beijing, Shanghai, Liaoning, Guangxi, Hunan, Guangdong, Tianjin, Sichuan, and Hubei Province. Furthermore, there were 0.18% of the hyper-aged type units located in Shanghai, Tianjin, and Liaoning Province.

In 2010, 79.4% of the units belonged to the aged type, covering 31 provinces, almost the entire of China. Among them, the aged type units in 27 provinces, except for Xinjiang, Tibet, and Qinghai, had reached more than 50%. There were 1% of units in Inner Mongolia, Shanghai, Jiangsu, Zhejiang, Shandong, Liaoning, Heilongjiang, Chongqing, and Sichuan Province that were the hyper-aged type.

In 2020, the aged and hyper-aged types were the predominant age structure types in China. The units of the aged type declined to 47.8%, whereas the hyper-aged type increased to 46.9%. The majority of the places in the northwest, northern, southern, and Yunnan and Guizhou Provinces in the southwest of China were the aged type. The northeast, eastern, central, and Chongqing and Sichuan in the southwest were mainly the hyper-aged type. Among them, more than 90% of the units in the northeast Heilongjiang, Jilin, and Liaoning Provinces were the hyper-aged type. The eastern rural areas were more advanced in their aging than the western rural areas. The units of young type declined to 5.3% and mainly concentrated in Tibet, Qinghai. This type was also sporadically distributed in southern Guangdong, eastern Fujian, central Ningxia, and western Sichuan.

Overall, rural population aging increased annually, and the regional disparity was significant. In comparison with 2000, the transition from the young type to the aged type was the most significant indication of how the RPA had risen to varied degrees and the age structure of the population had switched to the type with a higher aging rate. Furthermore, the range of the hyper-aged type of areas had been expanded.

### 4.2. Hot Spots Analysis of Rural Population Aging

The hot and cold spot analysis was performed to delineate the spatial cluster of rural population aging in China based on Getis–Ord Gi* statistics. The resultant z-score identified the states with high or low values of clustering spatially, as depicted in Figure 3a–c.

In 2000, significant hot spots (high cluster) of RPA were spread around the Shandong, Zhejiang, Shanghai, and Jiangsu Provinces in the eastern area, and there were small hot spots in: Fujian in the eastern area; Guangdong, Guangxi, and Hainan in the southern area; Henan and Hubei in the central area; and Sichuan and Chongqing in Southwest China. Significant cold spots (low cluster) were mostly spread across Gansu, Xinjiang, and Qinghai in the northeast area, Inner Mongolia in the northern area, Tibet in the southwest, and the Pearl River Delta agglomeration of southern Guangdong in Southern China.

The map for 2010 showed cold spots in Tibet, Xinjiang, Qinghai, Inner Mongolia, and the Pearl River Delta agglomeration in Southern Guangdong. Compared with 2000, cold spots in Tibet apparently expanded on a southern direction. Most of the hot spots were found in Chongqing and east of Sichuan in the southwest area, and Shandong, Jiangsu, Zhejiang in the eastern area, and Hunan in Central China. Some hot spots were also portrayed on the northeast of Inner Mongolia and Heilongjiang.

In 2020, the cold and hot spots were clearly divided into two sides, and the cold spots were biased toward the west, whereas the hot spots were biased toward the east of China. Notably, the cold spots in Inner Mongolia transitioned to hot spots, and the hot spots in Liaoning were enhanced. There was a slight decrease in the hot spots of RPA in Southern and Eastern China. The RPA distribution formed a significant low cluster pattern in Tibet, the west of Sichuan, and Qinghai meaning that it showed a low value. The low value is due to the harsh natural conditions and the complex and diverse climate terrain, making it a low value area for population density in China.

During the study period, the result of the analysis in China showed clear spatial patterns of RPA that the cold spots had significantly changed, mainly shown in the transition from being dispersed around the northwest to concentrated in Tibet and Qinghai. In addition, the RPA distribution formed hot spots that were increased in Chongqing and Sichuan in the southwest, Inner Mongolia, and Liaoning in the northeast of China.

### 4.3. Analysis of Directional Evolution of the RPA

In this section, the SDE (standard deviational ellipse) was utilized to analyze changes in the spatial evolution of RPA in China from 2000 to 2020. The SDE spatial evolution distribution of RPA is shown in Figure 4, and the variation of parameters is listed in Table 3. We also cited the Hu Line to analyze the changing characteristics of the center of gravity of the RPA. The Hu Line, starting at Heihe city in Heilongjiang province and ending with Tengchong city in Yunnan province, was consistent with the law of population distribution in China. The Hu Line is a line of comparison, proposed by the Chinese geographer Hu Huanyong in 1935, to divide China’s population density. In Hu’s opinion, on the southeast side of the Hu Line, 36% of the land area supports 96% of the population, whereas on the northwest side of the Hu Line, 64% of the land area accounts for about 4% of the population.

The area of the SDEs represents the concentration of RPA. During the study period, five-sixths of the ellipse area is to the southeast of the Hu Line because 96% of China’s population is in the southeast half of the Hu Line. The ellipse accounted for about one-third of China’s total land area, covering 26 provinces except Xinjiang, Tibet, Heilongjiang, Jilin, and Hainan, indicating that rural population aging was widely distributed (or dispersed) in China over time.

The mean centers of the SDEs captured the annual shifts in RPA. The mean centers formed in Henan Province, Central China, as they developed from the southeast to the northeast. However, this did not indicate that the rural population in Henan Province was the most severe. This was due to the fact that, in 2000, the RPA in the southern and eastern provinces such as Guangdong, Guangxi, Hunan, Jiangsu, and Shanghai was higher than other areas. However, in 2020, the RPA in Heilongjiang, Jilin, and Liaoning in Northeast China showed a high cluster, with more than 90% of the county administrative units being hyper-aged.

The azimuth ($tan_{\theta }/{}^{\circ}$) represents the orientation of the long axis of the SDEs, about which RPA concentrates. The azimuth varied between 32 and 34° from 2000 to 2020, although not significantly, indicating that RPA tends to orient around a northeast-to-southwest major axis. The short axis decreased, whereas the long axis increased, implying the aging population’s more pronounced directional characteristics.

Combined, these changes illustrate that rural population aging is increasing and becoming more concentrated toward the southwest and northeast of China, particularly to Chongqing and Sichuan in the southwest, and Heilongjiang, Jilin, and Liaoning in the northeast. This is contrary to the internal migration pattern that flows from north to south, indicating that although a multitude of workers from rural areas migrated from north to south, their parents (as well as their children) were still left behind in their hometowns, creating the so-called “empty hearted village” phenomenon.

### 4.4. The Driving Forces of RPA on Geographical Detector

There is significant regional variability in the population aging in rural China, according to the analysis of the hot spots and the directional distribution of RPA. This section discusses the findings of the GD (Geographical Detector) model in order to identify the main influencing factors of spatial and temporal differences in population aging in rural China.

#### 4.4.1. The Analysis of Factor Detector on RPA

To clarify the similarities and differences among the aging mechanisms in different regions of China, the factor detector calculated the q-values to represent the relative importance of potential factors in RPA (Table 4).

The proportion of people aged 55 to 64 (P55-64) and the longevity rate (LGV) were high *q-statistics*, indicating these two factors were the most important factors affecting rural aging in China. The initial aging level was the most important factor controlling the aging of the rural population in China, and the longevity of the elderly critically impacts rural aging. However, some aspects of healthcare and socioeconomics, such as the number of hospitals (HOS), number of beds (BED), and the per capita GDP (PGDP), showed low *q-statistics*. This was due to the fact that HOS and BED cannot reflect the medical staff’s diagnosis, treatment skills, and service attitudes towards patients. In addition, with the background of increasing aging in China, except for some of the most developed areas, particularly the Pearl River Delta agglomeration in the southern Guangdong Province, the PGDP plays a relatively unimportant role in rural population aging.

For different areas, the decisive factors were different. In Central, Eastern, Northeast, and Northwest China, the fertility rate (FER) showed a relatively strong determinant. One of the major demographic factors contributing to population aging is declining fertility. In Northeast China, the urbanization rate (UBZ), per capita years of education (PEDU), fertility rate (FER), longevity rate (LGV), and migration rate (MIG) were strong driving factors. Since the industrialization of the northeast began early, most of the agricultural population has shifted to the industrialized urban population. The higher the education level, the stronger the awareness of healthcare and the stronger the ability to be healthy. The northeast region is a historically multiethnic and culturally integrated region. Immigrant waves such as the “Rush to the Northeast” have greatly enriched the population gene pool of the northeast region and further increased the heterogeneity of the health status of the elderly population in the region. Rural areas in Northwest and Northern China are aging primarily as the result of demographic changes. In Southern China, the total population (TPOP) played a more significant role. As the most populous province, Guangdong has plenty of young and middle-aged migrant workers, making the problem of population aging less severe. Furthermore, the not-aged population moving into the city has a “diluting effect” on the aging of the population, and the aggregation and diffusion effects caused by different stages of city development are important for both aging migration.

#### 4.4.2. The Analysis of Interaction Detector on RPA

In total, 55 pairs of interactions were calculated between 11 variables using the interaction detector. Table 5 shows that the interaction relationships among all the demographic, socioeconomic, healthcare, and educational factors. The *q statistics* in the table represent the explanatory power of the two factors acting together on the RPA. The synergistic effects between each pair of driving factors were manifested as bivariate- enhanced or nonlinear-enhanced influences on RPA in this study. This indicates that the interaction between the two driving factors had a stronger influence than each individual factor on the RPA. Among the interactions of all factors, the longevity rate (LGV), intersected with the other indicators, yielded the strongest value, with all reaching 0.72 or more, indicating that the longevity rate (LGV) was the most predominant factor affecting the RPA. The $q\;\left(\mathrm{LGV}\cap P55\sim 64\right)$ was the maximum (0.82), indicating that the interaction between longevity rate and P55-64 was the strongest. In addition, among the interactions of the socioeconomic factors, $q\;\left(TPOP\cap P55\sim 64\right)$ was the maximum (0.61).

#### 4.4.3. Statistical Significance of Differences among Driving Factors

The significance of varying influence among the 11 factors was examined via the ecological detector and risk detector. Table 5 shows that upward of half were statistically significant, whereas statistically significant differences existed between the urbanization rate (UBZ) and the fertility rate (FER), the numbers of beds (BED) and the total population (TPOP), the per capita years of education (PEDU) and total population (TPOP), and the numbers of bed (BED) and per capita years of education (PEDU). Combining the results of the factor detector, it can be concluded that the fertility rate (FER) had a greater impact on RPA compared to urbanization (UBZ); and the influence of per capita years of education (PEDU) on RPA was significantly stronger than that of the number of beds (BED).

## 5. Discussions

In this paper, we focused on analyzing the spatial temporal patterns and potential influence factors of rural population aging in China from 2000 to 2020. The results revealed that the population aging in rural China is characterized by significant spatial heterogeneity. First, we analyzed the hot (high cluster) and cold spots of rural population aging by Getis-Ord Gi*. The analysis found that three major hot spots were formed in the northeast (northeast of Inner Mongolia and Liaoning), eastern (Zhejiang and Jiangsu), and southwest area (Chongqing and Sichuan), whereas cold spot clusters were formed in Tibet, Qinghai, the southern Guangdong, and the western Sichuan Province. Second, the standard deviational ellipse was utilized to analyze the spatial directional evolution of rural population aging. It was discovered that the aging of rural population tends to orient around a northeast-to-southwest major axis, implying that the southwest and northeast face more challenges regarding aging. Finally, the potential influence factors of rural population aging in China were detected. The longevity rate, fertility rate, and proportion of aged 55 to 64 years old are predominant factors of rural population aging in China.

The results of this study need to be considered in combination with those from other recent studies that provide spatiality and the driving forces of population aging in China [5,6,8,40]. Our results preliminarily summarized the characteristics of different types of aging and highlighted changes in the spatiality of rural population aging, generally presenting a southwest-to-northeast evolution pattern. In addition, we concluded that the predominant influence factors of rural population aging in China are mainly caused by the natural changes in population age structure, and socioeconomic factors are the important reasons for the differences in rural population aging in different regions. The findings indicate that population aging is not only a spatial–temporal process but also a natural population structure change process.

There are some deficiencies in this study to be discussed. First, the county-level analysis units are rather large to measure accurately the population aging in towns and villages. Although the county-level panel data used in this study reflected the spatial–temporal distribution and evolutionary patterns of China’s rural population aging from a macroscopic perspective, the distribution of older populations in a county from a microscopic perspective cannot easily be reflected. Even so, due to the availability of data, 2846 counties in 31 provinces on mainland China were still used as the unit of analysis. Second, the selection and discussion of variables influencing the spatial and temporal patterns are insufficiently exhaustive. Noneconomic factors such as natural environmental conditions, diet structures, and personalities should be further discussed. Third, there is also limited demonstration regarding how the aging of the rural population differs in different regions. Take Shenzhen and Dongguan cities in Guangdong Province, for example. Due to the highly developed socioeconomics, relatively open household registration system, and many employment opportunities, young laborers continuously immigrate and earn a living there, making them two of the youngest cities in Guangdong Province and even in China. Therefore, future research should concentrate on the heterogeneous factors influencing rural population aging at a local level to provide a reference for the development of more specific solutions to the rural aging problem.

## 6. Conclusions

The study results revealed the spatial-temporal distribution and predominant factors of population aging in rural China from 2000 to 2020, and found obvious differences in the aging patterns within each regions. Consequently, it is necessary for the Chinese Government to make scientific and precise policy decisions with full consideration of the spatial variation. First, retirement resources and preferential policies should be appropriately tilted. For example, according to the needs of rural areas, a number of nursing homes should be planned and built in places close to township health centers to provide centralized care for the disabled and elderly. Moreover, well-being for migrant workers in rural areas needs to be elevated to ensure that they have the same level of social welfare as urban residents. Health-related infrastructures, such as hospitals, care centers, and nursery centers need to be constructed, especially in the less developed northern areas of China, to improve the accessibility of the aged to healthcare. As the number of rural nuclear families and empty nesters living alone increases, the ability of families to age in place diminishes, causing the tradition of relying on children to age in place to change [41]. Therefore, in the future, there will be an urgent need to introduce social capital, i.e., government and enterprise cooperation in providing social elderly care services, to solve part of the rural elderly care problem. Government subsidies and tax breaks can be used to encourage social forces to set up private institutions for the elderly. In addition, the private sector should provide services that enrich learning, recreation, and leisure activities for the rural elderly. Furthermore, private senior care institutions should also focus on strengthening their branding and determining a business model that fits their needs, taking into account the elderly consumer, relevant government policies and measures, and their own business philosophy to adopt a series of suitable business methods.

## Figures and Tables

**Figure 1 ijerph-19-15631-f001:**
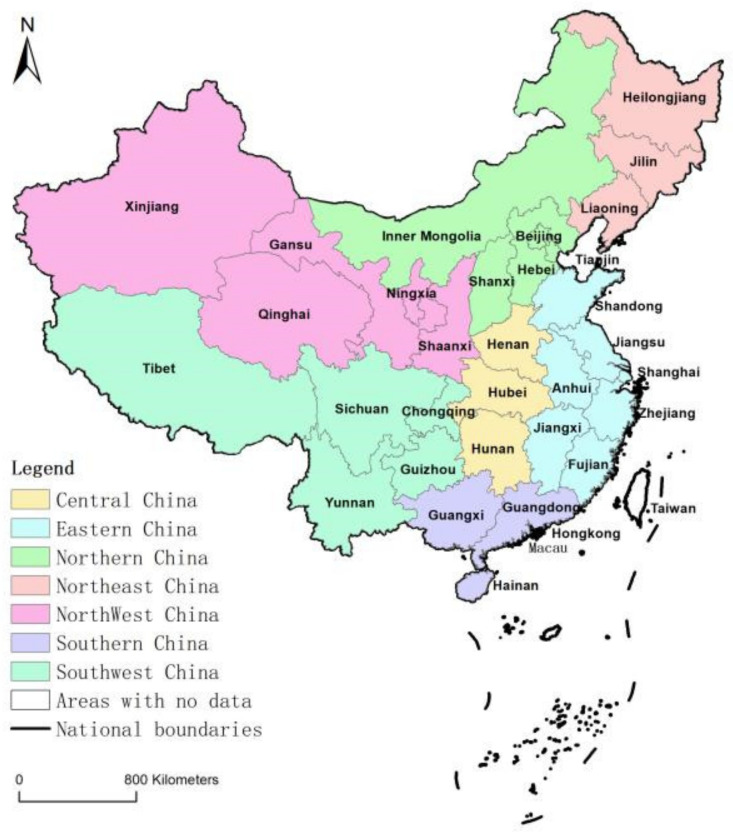
Agglomeration of provincial level administrative units.

**Figure 2 ijerph-19-15631-f002:**
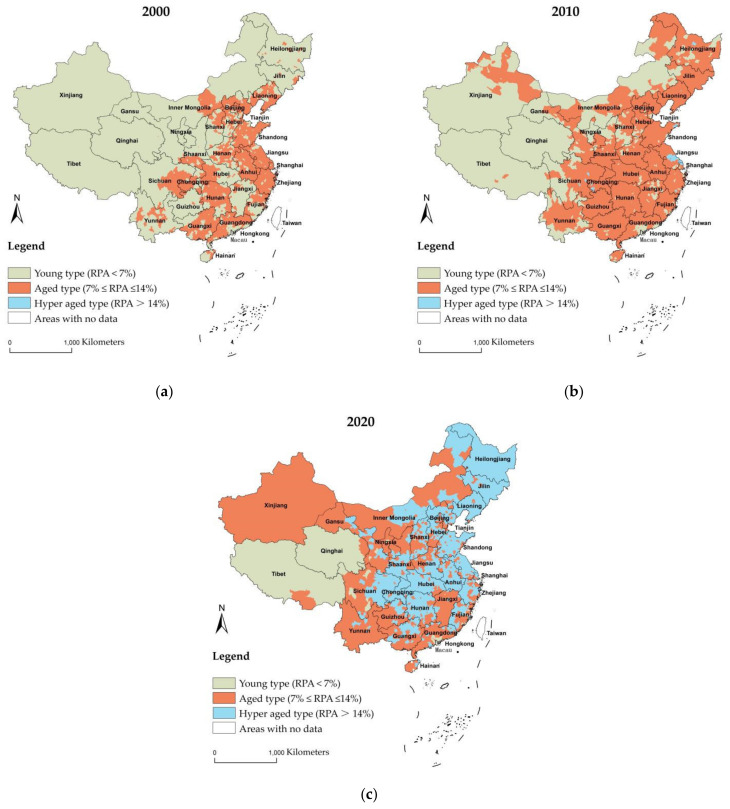
Spatial distribution profiles of China’s rural population aging ratio in 2000 (**a**), 2010 (**b**), and 2020 (**c**).

**Figure 3 ijerph-19-15631-f003:**
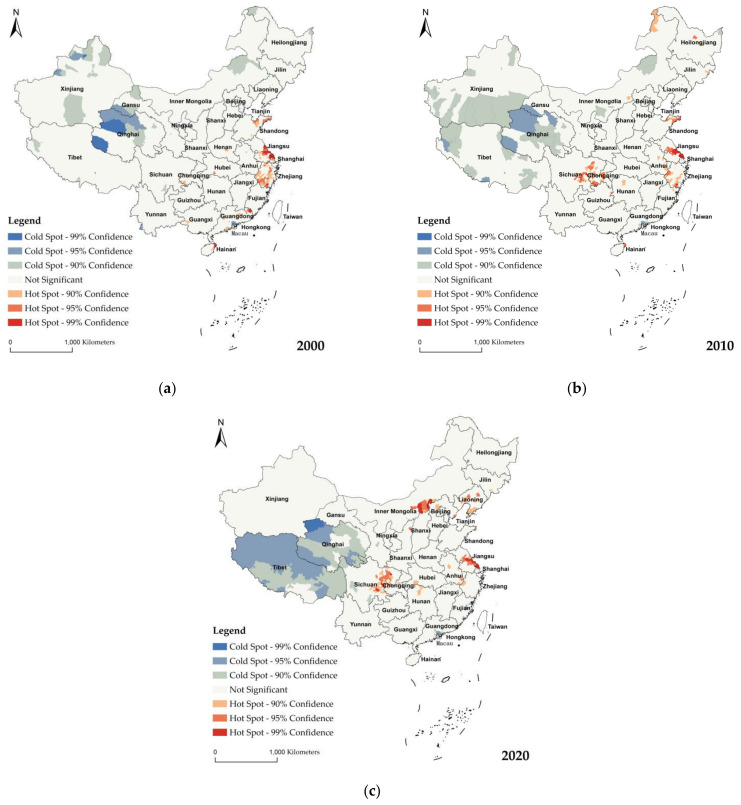
Spatial clustering (hot and cold spots) analysis of China’s rural population aging in 2000 (**a**), 2010 (**b**), and 2020 (**c**).

**Figure 4 ijerph-19-15631-f004:**
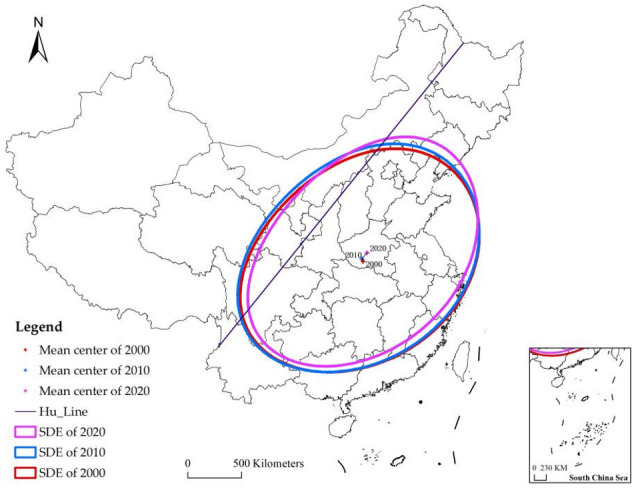
Directional distribution of rural population aging from 2000 to 2020.

**Table 1 ijerph-19-15631-t001:** Description of influencing variables on RPA.

Variable Systems	Variables (Abbreviation)	Descriptions
Natural and mechanical demographic characteristics	Total population (TPOP)	The total population in each county
	The proportion of 55 to 64 years old (P55-64) ten years ago	P55-64 represents the base of aged people
	Fertility rate (FER)	The proportion of births in the total population
	Longevity rate (LGV)	The proportion of the population aged 80 and above in the total population
	Migration rate (MIG)	Hukou^1-^registered population in other places/Hukou-registered population in local places
Socioeconomic characteristics	Per capita GDP (PGDP)	The per capita GDP of each county
	Urbanization rate (UBZ)	The proportion of nonagricultural Hukou in total population
Healthcare accessibility characteristics	Number of hospitals (HOS)	The number of hospitals in each county
	Number of beds (BED)	The number of beds in each county
Educational characteristics	Per capita years of education (PEDU)	Refers to the average years of education of the population aged 6 and over. A college degree or above is calculated as 16 years, 12 years for high school, 9 years for junior high school, 6 years for primary school, and 0 years for illiteracy
	Illiteracy rate (ILT)	The proportion of illiterate people aged 15 and over in the total population

Hukou^1^ means the household registration system in China, which includes nonagricultural Hukou and agricultural Hukou.

**Table 2 ijerph-19-15631-t002:** The interaction types of two factors and the interactive relationship.

Title 1	Description
Weakened, nonlinear	$q(x_{1}\cap x_{2})<Min\;\left(q(x_{1}\right),\;q(x_{2}))$
Weakened, univariate	$Min\;\left(q(x_{1}\right),\;q(x_{2}))<q(x_{1}\cap x_{2})<Max\;\left(q(x_{1}\right),\;q(x_{2}))$
Enhanced, nonlinear	$q(x_{1}\cap x_{2})>q(x_{1})+q(x_{2})$
Enhanced, bivariate	$q(x_{1}\cap x_{2})<Max\;\left(q(x_{1}\right),\;q(x_{2}))$
Independent	$q(x_{1}\cap x_{2})=q(x_{1})+q(x_{2})$

**Table 3 ijerph-19-15631-t003:** Variation in parameters of the SDE of RPA in China from 2000 to 2020.

Year	Center X	Center Y	$tan_{\theta }/{}^{\circ}$	$\mathrm{Long}\;\mathrm{Axis}\;(\alpha_{x})/\mathrm{km}$	$\mathrm{Short}\;\mathrm{Axis}\;(\alpha_{y})/\mathrm{km}$
2000	32.91°N	112.71°E	69.5°	12,776.79	8439.85
2010	33.12°N	112.61°E	68.8°	12,962.98	8589.18
2020	33.59°E	113.19°E	63.6°	12,839.65	8140.22

**Table 4 ijerph-19-15631-t004:** The *q-statistics* of the factor detector in various regions.

Factors	Central	Eastern	Northern	Northeast	Northwest	Southern	Southwest
TPOP	0.057	0.467	0.125	0.516	0.087	0.01	0.113
P55-64	0.052	0.672	0.273	0.722	0.138	0.066	0.023
FER	0.082	0.516	0.104	0.629	0.09	0.131	0.177
LGV	0.383	0.427	0.619	0.814	0.369	0.081	0.707
MIG	0.175	0.651	0.364	0.486	0.118	0.048	0.159
PGDP	0.165	0.711	0.065	0.644	0.157	0.078	0.127
UBZ	0.323	0.893	0.372	0.798	0.153	0.104	0.351
HOS	0.055	0.039	0.104	0.123	0.025	0.141	0.435
BED	0.015	0.054	0.088	0.229	0.117	0.146	0.368
PEDU	0.04	0.065	0.167	0.621	0.105	0.125	0.458
ILT	0.028	0.124	0.205	0.242	0.094	0.089	0.388

**Table 5 ijerph-19-15631-t005:** The interaction detector analysis and statistically significant differences of the driving factors on RPA.

Variables	Demographic					Socioeconomic		Healthcare		Educational	
	TPOP	P55-64	FER	LGV	MIG	PGDP	UBZ	HOS	BED	PEDU	ILT
TPOP											
P55-64	0.61 (EB, Y)										
FER	0.25 (EB, Y)	0.61 (EB, Y)									
LGV	0.74 (EB, Y)	0.82 (EB, Y)	0.74 (EB, Y)								
MIG	0.20 (EB, Y)	0.61 (EB, Y)	0.18 (EN, Y)	0.74 (EB, Y)							
PGDP	0.19 (EN, Y)	0.62 (EB, Y)	0.14 (EB, Y)	0.73 (EB, Y)	0.11 (EN, Y)						
UBZ	0.24 (EB, Y)	0.62 (EB, Y)	0.20 (EB, N)	0.73 (EB, Y)	0.15 (EN, Y)	0.14 (EN, Y)					
HOS	0.25 (EB, Y)	0.62 (EB, Y)	0.30 (EN, Y)	0.72 (EB, Y)	0.24 (EB, Y)	0.24 (EN, Y)	0.26 (EB, Y)				
BED	0.20 (EB, N)	0.61 (EB, Y)	0.26 (EB, Y)	0.73 (EB, Y)	0.21 (EB, Y)	0.21 (EN, Y)	0.26 (EB, Y)	0.25 (EB, Y)			
PEDU	0.24 (EB, N)	0.64 (EB, Y)	0.26 (EB, Y)	0.74 (EB, Y)	0.20 (EB, Y)	0.2 (EN, Y)	0.22 (EB, Y)	0.28 (EB, Y)	0.25 (EB, N)		
ILT	0.22 (EB, Y)	0.64 (EB, Y)	0.22 (EB, Y)	0.74 (EB, Y)	0.19 (EN, Y)	0.19 (EN, Y)	0.23 (EB, Y)	0.27 (EB, Y)	0.24 (EB, Y)	0.20 (EB, Y)	

Note: (EN) represents the nonlinear enhancement of two factors, and (EB) represents the binary enhancing of two factors (see Table 2); Y represents the risk difference between the two factors is significant with confidence of 95%, and N represents no significant difference.

## Data Availability

No new data were created or analyzed in this study. Data sharing is not applicable to this article.

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
