# Peer review of "Spatial–Temporal Patterns of Population Aging in Rural China"

_ijerph, 2022, doi:10.3390/ijerph192315631_

Round 1

Reviewer 1 Report

Comments and suggestions for authors are in attachment. 

Reviewer 2 Report

The challenge that the increase in the number of elderly people in the world is going to pose is one of the most relevant political issues of this century. This phenomenon is going to be very relevant in China, as it goes hand in hand with a drop in the birth rate for years. Therefore, the topic of the paper seems very relevant to me.

Lines 79-90. This information should not appear in the introduction. This information is from the “method” section, therefore it must be included in this section.

On the other hand, at the end of the introduction, the objective must be clearly written. For e.g. following the formula “The aim of this paper…”

Therefore, I suggest authors clearly write the objective. And it should appear at the end of the introduction.

Lines 122-123. The paper is about research, therefore, this formula is well known in the scientific field, it is not necessary to appear in the paper.

Lines 176-177. In my opinion, I believe that it is not correct to use the term "independent variables" This study is not experimental, it is a selective correlational study, therefore, causal relationships cannot be established, only correlations can be established, but correlation exists does not necessarily mean that there is causality. I suggest using the term "variables" without any adjectives.

The figures (fig 2 and fig 3) of the paper are very interesting, but they have a very small format, they are barely visible. Therefore, I suggest they be laid out in a larger format.

The information in Table 4 is very interesting, but it would be clearer in another format. I suggest the authors consider writing the regions of China in the first row and the variables in the first column. That is, modify the orientation of the table.

I suggest authors merge Table 5 and Table 6 into a single table.

The conclusions (line 436) should be after the discussion (line 470)

Conclusions should be concise and should be “to the point”. Therefore delete from line 437 to line 440.

Order the rest of the conclusions by summarizing them.

Author Response

(PS: We have provided a version of the revised manuscript where all changes are tracked (in addition to a clean version. )

Point 1: The challenge that the increase in the number of elderly people in the world is going to pose is one of the most relevant political issues of this century. This phenomenon is going to be very relevant in China, as it goes hand in hand with a drop in the birth rate for years. Therefore, the topic of the paper seems very relevant to me.

Response 1: We thank you for these supportive words and your constructive comments.

Point 2: Lines 79-90. This information should not appear in the introduction. This information is from the “method” section, therefore it must be included in this section.

Response 2: We have moved lines 79-90 to the “Methods” section and revised the statement of the purpose of the study in the “Introduction” section. Please see Lines 192-198 in the revised version.

Point 3: On the other hand, at the end of the introduction, the objective must be clearly written. For e.g. following the formula “The aim of this paper…”

Response 3: We have added a statement of the aim of the study in the introduction: “This study set out to examine the spatial heterogeneity and influencing factors of population aging in rural China from 2000 to 2020 with county-level panel data. Based on a large spatial scale and time span, we comprehensively use GIS analysis tools to study the spatial and temporal distribution law and influencing factors of China's rural population aging. The aims of this study are to reveal the comprehensive deepening law of rural population aging and provide a scientific basis for active aging development and population policy adjustment in China.”

Point 4: Lines 122-123. The paper is about research, therefore, this formula is well known in the scientific field, it is not necessary to appear in the paper.

Response 4: The formula has been deleted.

Point 5: Lines 176-177. In my opinion, I believe that it is not correct to use the term "independent variables" This study is not experimental, it is a selective correlational study, therefore, causal relationships cannot be established, only correlations can be established, but correlation exists does not necessarily mean that there is causality. I suggest using the term "variables" without any adjectives.

Response 5: Corrected.

Point 6: The figures (fig 2 and fig 3) of the paper are very interesting, but they have a very small format, they are barely visible. Therefore, I suggest they be laid out in a larger format.

Response 6: Larger formats for figure 2 and figure 3 have been laid out.

Point 7: The information in Table 4 is very interesting, but it would be clearer in another format. I suggest the authors consider writing the regions of China in the first row and the variables in the first column. That is, modify the orientation of the table.

Response 7: Done.

Point 8: I suggest authors merge Table 5 and Table 6 into a single table.

Response 8: Done. Table 6 has been deleted and grouped into Table 5.

Point 9: The conclusions (line 436) should be after the discussion (line 470)

Response 9: The article was reformatted according to the content and we decided to integrate the Discussions and Conclusions sections together under the name "Discussions and Conclusions".

Point 10: Conclusions should be concise and should be “to the point”. Therefore delete from line 437 to line 440.

Response 10: Done.

Point 11: Order the rest of the conclusions by summarizing them.

Response 11: Corrected. Please see Lines 491-555 in the revised version.

Reviewer 3 Report

The aim of  the article “Spatial-temporal patterns of population aging in rural China” is to examine the spatial heterogeneity and influencing factors of  population aging in rural China from 2000 to 2020 with county-level panel data. The method was used to solve the research problems: Getis-ord Gi*, the Standard Deviational Ellipse method, the Geographical Detector method. The adopted research method is correct.

The results of the study were presented in a consistent manner, supplemented with figures to illustrate the analyzed problems. The conclusions result from the conducted analysis, they are complete. The conclusions are very clear, pointing to specific solutions. The type and scope of source materials is adequate to the analyzed problem. The article has the correct structure.

Summing up, it should be stated that the article is coherent and logical. Moreover, the article formulates further directions of research on the discussed issue. In my opinion, it will be an interesting development and supplement to the topic of Spatial-temporal patterns of population aging in rural China.

I noticed an error: the difference between the percentages is percentage points (p.p.): line 10, is “6.61%”; should be “6.61 p.p.

Author Response

Point 1: The aim of the article “Spatial-temporal patterns of population aging in rural China” is to examine the spatial heterogeneity and influencing factors of population aging in rural China from 2000 to 2020 with county-level panel data. The method was used to solve the research problems: Getis-ord Gi*, the Standard Deviational Ellipse method, the Geographical Detector method. The adopted research method is correct.

Response 1: We appreciate your affirmation of the methods used in this study.

Point 2: The results of the study were presented in a consistent manner, supplemented with figures to illustrate the analyzed problems. The conclusions result from the conducted analysis, they are complete. The conclusions are very clear, pointing to specific solutions. The type and scope of source materials is adequate to the analyzed problem. The article has the correct structure.

Response 2: We thank you again for these supportive words.

Point 3: Summing up, it should be stated that the article is coherent and logical. Moreover, the article formulates further directions of research on the discussed issue. In my opinion, it will be an interesting development and supplement to the topic of Spatial-temporal patterns of population aging in rural China.

Response 3: Thank you very much.

Point 4: I noticed an error: the difference between the percentages is percentage points (p.p.): line 10, is “6.61%”; should be “6.61 p.p.

Response 4: Corrected.